# A Novel Approach to Multi-Provider Network Slice Selector for 5G and Future Communication Systems

**DOI:** 10.3390/s22166066

**Published:** 2022-08-13

**Authors:** Douglas Chagas da Silva, José Olimpio Rodrigues Batista, Marco Antonio Firmino de Sousa, Gustavo Marques Mostaço, Claudio de Castro Monteiro, Graça Bressan, Carlos Eduardo Cugnasca , Regina Melo Silveira

**Affiliations:** 1Department of Computer Engineering and Digital Systems, Escola Politécnica, University of São Paulo, São Paulo 05508010, Brazil; 2Department of Information Systems, Campus Palmas, State University of Tocantins, Palmas 77020122, Brazil; 3Federal Institute of Tocantins, Computer Networks Research Group, Palmas 77021090, Brazil

**Keywords:** 5G, beyond 5G, networks softwarization, multi-criteria decision methods, Network Slice Selection Function (NSSF)

## Abstract

The Network Slice Selection Function (NSSF) in heterogeneous technology environments is a complex problem, which still does not have a fully acceptable solution. Thus, the implementation of new network selection strategies represents an important issue in development, mainly due to the growing demand for applications and scenarios involving 5G and future networks. This work presents an integrated solution for the NSSF problem, called the Network Slice Selection Function Decision-Aid Framework (NSSF DAF), which consists of a distributed solution in which a part is executed on the user’s equipment (for example, smartphones, Unmanned Aerial Vehicles, IoT brokers) functioning as a transparent service, and another at the Edge of the operator or service provider. It requires a low consumption of computing resources from mobile devices and offers complete independence from the network operator. For this purpose, protocols and software tools are used to classify slices, employing the following four multicriteria methods to aid decision making: VIKOR (Visekriterijumska Optimizacija i Kompromisno Resenje), COPRAS (Complex Proportional Assessment), TOPSIS (Technique for Order Preference by Similarity to Ideal Solution) and Promethee II (Preference Ranking Organization Method for Enrichment Evaluations). The general objective is to verify the similarity among these methods and applications to the slice classification and selection process, considering a specific scenario in the framework. It also uses machine learning through the K-means clustering algorithm, adopting a hybrid solution in the implementation and operation of the NSSF service in multi-domain slicing environments of heterogeneous mobile networks. Testbeds were conducted to validate the proposed framework, mapping the adequate quality of service requirements. The results indicate a real possibility of offering a complete solution to the NSSF problem that can be implemented in Edge, in Core, or even in the 5G Radio Base Station itself, without the incremental computational cost of the end user’s equipment, allowing for an adequate quality of experience.

## 1. Introduction

Convergence among networks of different technologies has become a reality. The processing power of mobile devices and the diversity of services that can be used with them guide the way that access technology infrastructures are modeled. The term convergence hence refers to the possibility of providing user access through different technologies, enabling the use of various services such as voice, video, and data in general [1].

The application of the Network Slicing (NS) concept provides an avenue to new mobile network solutions, including 5G networks (Fifth Generation Mobile Networks) and future communication systems due to stricter requirements in terms of their Key Performance Indicators (KPIs)—isolation, latency, mobility, peak data rate, and so on, as shown in Figure 1. Although several proposals point out paths in domains that involve heterogeneous technologies, it is still not possible to aggregate several functionalities in a single and fully functional approach to set the operation and management mechanisms of each slice, or to provide subsidies for scalability, orchestration, and support for decision making [1,2].

The task of selecting a Radio Access Network (RAN) in an environment of heterogeneous technologies is difficult since operators can provide specific types of slices directly to meet the requirements of an application or multiple slices for the requirements of the same user. Therefore, there is still no solution or technique that understands all the aspects and mechanisms of access to these technologies [3,4]. The implementation of new selection techniques becomes necessary due to the demand in the growing use of vehicular networks, patient monitoring, smart cities, and Internet of Things (IoT), among other technologies and scenarios involving network convergence, mobility management, and service continuity in 5G networks and beyond [5,6,7,8,9,10,11].

The main issues arising from this process are as follows: How to choose the best slice? Do the selected slices provide the necessary requirements for the user? Are the selection criteria modeled in a generic way, independently of the access technology?

In this work, a study was conducted on the use of computational tools for classifying slices, to support the decision-making process using aids for our proposed Network Slice Selection Function Decision-Aid Framework (NSSF DAF). Four strategies were suggested, namely, VIKOR (Visekriterijumska Optimizacija i Kompromisno Resenje), COPRAS (Complex Proportional Assessment), TOPSIS (Technique for Order Preference by Similarity to Ideal Solution) and Promethee II (Preference Ranking Organization Method for Enrichment Evaluations) [13].

The general objective was to verify the similarity among these methods and applications in the slice classification and selection processes considering a specific scenario and then propose the framework that provides the best experience to the user. Machine Learning (ML) with the K-means clustering algorithm was selected, adopting a hybrid solution to the implementation and operation of the Network Slice Selection Function (NSSF) service in a multi-domain slicing environments of heterogeneous mobile networks. To validate the framework, testbeds were conducted to map the necessary Quality of Service (QoS) requirements.

Thus, a new approach that employs techniques aimed at the integration and interoperability between RAN or Open Radio Access Network (O-RAN), Edge and Core networks, based on an efficient and robust Slice Selection service (virtual network selection), under an architecture that provides compatibility with the standards specification in progress, is a promising solution for 5G mobile networks. It is also applicable to various segments, such as the integration of a large amount of data from devices linked to the IoT context (e.g., smart homes, patient monitoring, Wireless Sensor Networks—WSNs) with cloud services; integration of applications that have multimedia requirements, high density of video and audio traffic, such as applications that use Virtual Reality (VR) and Augmented Reality (AR) technologies; provision of specialized networks with sophisticated slice selection and security mechanisms to provide services for autonomous vehicle networks in different models and topologies (e.g., V2V—Vehicle-to-Vehicle, V2I—Vehicle-to-Infrastructure, V2X—Vehicle-to-Everything) [14,15,16].

More than the comprehension of NS, this work contributes to improving the network slice selection process. A framework that implements the NSSF service in vertical and horizontal models is proposed, where the handover decision is shared between the network edge and the User Equipment (UE) at network runtime.

The remainder of this paper is organized as follows: Section 2 details its main contributions; Section 3 contextualizes the related work; Section 4 exposes the general problem and the need to use edge computation in the slice selection model; Section 5 presents the proposed solution framework for Network Slicing selection; Section 6 describes the scenario and the methodology used in the experiments, as well as reports and discusses the results obtained. Finally, Section 7 concludes the work.

## 2. Main Contributions

Specifically, the points presented below are original contributions to this paper:(a)Proposal of a framework that implements the NSSF service in vertical and horizontal models, where the handover decision is shared between the network edge and the user’s equipment on network runtime;(b)Suitability of NSSF function originally from 5G core to network edge, to UE, or to 5G Radio Base Station (gNodeB). The deployment model is defined according to the institutional operating scenario;(c)Slice selection model implementation based on multicriteria decision techniques and machine learning, using hybrid algorithms;(d)Strategy formulation independent of packet marking type (e.g., SR-IPv6, MPLS, VXLAN, VPN, and GTP) for analyzing data flows and forwarding to available slices;(e)Defining approach for collecting QoS and Quality of Experience (QoE) metrics directly from the TCP/IP stack, and signaling the UE profile, without legacy infrastructure modifications;(f)Definition of network datasets characterization and acquisition models, in addition to decision matrices assembly in network runtime, using data analytics tools;(g)Integrated model specification and implementation with market orchestration tools such as ONAP, OSM, and OpenNESS;(h)Slice selection embedded application development for multistream mobile devices.

## 3. Related Work

The purpose of this section is to contextualize the slice selection problem in the scenario of heterogeneous mobile networks (e.g., 5G and Wi-Fi 6) to support the expected contribution of this work in Section 5.1.

Several solutions have been proposed to evaluate and select the best slice, considering the QoS requirements for telephony and data services. In general, the literature suggests approaches that consider the following situation: Given a set of criteria or network parameters, the works verify at any given time and among the available slices, which better fits the user needs, supporting network exchange (handover process) for the mobile [4,5,17,18]. In this case, the process of choosing the slice is subject to a number of criteria.

A higher number of mobile devices, as well as the variety of services in the vertical slicing model, require the optimized development of an appropriate logical architecture, which allows scalability, energy efficiency, simplification of network functions, and provides a business model (CAPEX and OPEX) that uses the computational infrastructure in operation. Thus, the Network Slicing architecture, with a well-defined Slice Selection service, emerges as the main solution for next-generation mobile networks [5,19,20].

Most papers focused on strategies that offer telecommunications operators mechanisms that can provide and control resources for each virtual instance. This includes customization to meet the specific requirements of users in a pre-defined structure of the allocation of resources, that is, without the decision of the choice of the slice being passed by the end-user, and ignoring the type of RAN to which it is linked.

These points raise interesting questions that need to be evaluated, such as the End-to-End (E2E) model in NS environments; the guarantee of continuity of services in a roaming NS architecture without the interruption of active services and with guaranteed stability in the delivery of data in scenarios with medium and high mobility; interoperability between different network slice architectures under different administrative domains; security in the transmission of sensitive data; and the migration of all physical network functions to logical networks (Network Function Virtualization—NFV) abstractions [1,3,16,21].

According to [1,22,23,24], the issues listed above are important and have motivated several research laboratories and institutions to seek to standardize the strategies and solutions pointed out so far to create standards for the industry. Those institutions and working groups include but are not limited to the Internet Engineering Task Force (IETF) [25], Next Generation Mobile Networks (NGMN) [26], Open Networking Foundation (ONF) [27], 3rd Generation Partnership Project (3GPP) [28], and the European Telecommunications Standards Institute (ETSI) [29]. Therefore, new proposals of approaches, as well as modifications to existing architectures, constitute an open field from the point of view of researchers and the market [5].

An issue that has arisen in some countries is whether the slicing of the 5G network will be consistent with the network neutrality regulations. Some say that the practical implications for current open Internet rules are speculative at this stage concerning 5G. That is because the evolution of the different 5G elements, such as NS, depends not only on the occasional technological capabilities but also on the market demand, the degree of competition, the commercial strategies, and so on [30].

There are several contributions in the literature for admission control, resource allocation, and billing mechanisms in virtualized wireless networks [31,32]. However, the discussion remains open in the scientific community in terms of automated mechanisms for slicing and monetization in 5G [33].

Relevant scientific work performed on NSSF is discussed in the following and summarized in Table 1.

In addition to the Virtual Network Function (VNF) for selecting slices, Rivera, Khan, Mehmood and Song [34] argue that the literature shows that a management system is also required to provide, update, and control the VNFs of the physical layer that makes up the slices. They focused on the provision and deployment of Data Plane Network Slices and the selection of a proper slice by taking identification parameters from the user equipment. They also confirmed that traffic control rules can be deployed with minimal resources without affecting the efficiency of the Data Plane slices. As the proposed system evolves, a new degree of automation can be developed with a monitoring agent that can supervise the status of SPGW-U traffic in real-time.

The works in [8,35] proposed NSSF based on Technique for Order Preference by Similarity to Ideal Solution (TOPSIS). According to the principles of network division, a mobile user/terminal may choose from several connectivity alternatives available based on criteria related to slicing performance, service requirements, and user preferences. NSSF is perceived as a logical evolution from the ABC concept to 5G and beyond mobile systems. NSSF is modeled as a Multiple-Criteria Decision-Making (MCDM) problem exploring the principle of TOPSIS for classifying the available slices based on their attributes and weights. TOPSIS is a widely accepted decision-making tool, considering its understandable logic, algorithmic logic, and mathematical form. However, it fails to provide consistent results due to the phenomenon of rank reversibility.

As for the TOPSIS methods, the results in [8] demonstrated that the standard deviation weighting technique (SD-TOPSIS) presents significantly better performances regarding rank reversibility, but it has poor performance in terms of complexity. In this sense, they suggested applying the entropy weighting technique (E-TOPSIS), especially when faced with characteristics such as the fine granularity of slices. The authors also concluded that the improvements in TOPSIS methods are closely related to alternative methods for normalization and ranking phases, especially when managing practical implementation situations. As for future work, the intention is to analyze the influence of several alternative standardization techniques and classification methods on the NSSF performance.

Besides, the results from [35] consider the three stages of the decision-making process (normalization, weighting, and ranking), and show that proposed alternative techniques, such as linear normalization (MAX-MIN), the weighting of variance, and binary classification alternatives, presented a positive influence by significantly reducing classification reversibility and computational complexity among the tested scenarios. This justifies the need to consider MCDM methods as a potential solution to the network slice selection problem in 5G mobile systems and their future generations. As for future work, the performance of other MCDM algorithms should be analyzed in terms of ranking reversibility.

The work in [36] presented a new approach that aimed at increasing service utilization and the efficiency of network slices, since slice selection is one of the key elements of 5G Network Slicing. In the proposed method, each network slice is considered as a different service and is represented by a website system with its own database. Then, the GET method is used to link the user equipment to multiple systems at the same time, focusing mainly on passing the required parameters by URL. By using this approach, Multiple-Service UE will be able to connect simultaneously to several networks, obtaining a session on the related networks for a specific time. As future work, the authors suggested providing the user with a PURE connection to these networks, meaning that by passing the parameters to the new system, the users can be registered to all existing systems, benefiting from their full capabilities, instead of having a temporary connection as observed in the present work.

Dimolitsas [37] presented a multicriteria decision framework for the optimal selection of Edge Points of Presence (EPoPs) to deploy a network slice. The EPoP Selecting Framework is composed of the following three main components: (i) the Service Registry, which contains the required KPI values for each EPoP candidate; (ii) the Filtering Engine, which is responsible for the initial filtering of the candidate EPoPs according to the user’s requirements; and (3) the PoP Ranking Mechanism, which selects the most appropriate EPoP for slice deployment based on the multi-criteria Fuzzy Analytic Hierarchy Process (FAHP) method. The proposed framework was assessed under a realistic scenario in comparison with simple filtering and the single-object (FAHP) approach. The results indicate the relevance of the proposed two-stage method in meeting the user’s requirements for hardware and software, allowing for the communication between slices and optimal resource allocation from the provider’s point of view. Future work should include the implementation of a distributed selection mechanism to improve service discovery and communication between slices, hence reducing the cost of deployment.

The work in [38] focuses on maximizing system resource utilization while guaranteeing the satisfaction degree among users by exploring the E2E network slicing problem. Using theoretical analysis, the authors proved that this is a NP-hard problem, and then they proposed a Genetic Algorithm (GA) to solve the optimization problem. A simulation experiment was conducted to validate the proposed GA algorithm, showing that this method obtained better access and transmission performance when compared to traditional selection methods based on the Received Signal Strength (RSS) or greedy algorithms. The authors did not state future work intentions, but it is reasonable to suggest the conduction of real-world experiments using the proposed GA algorithm to validate the simulation results from this work.

Silva et al. [39] presented a solution for 5G network slice selection in IoT scenarios. It uses Edge Computing resources with the application of hybrid machine learning algorithms and MCDM methods to permit IoT applications to better adapt data processing and routing, providing a better experience for users. From the results, the proposed solution proved to be efficient and the adopted MCDM methods show a similar performance. This demonstrates the high level of flexibility in the ranking of alternatives for the proposed methods as a function of the adopted weights. The authors state that in future work it is necessary to consider different scenarios for new experiments, in addition to further development in the proposed algorithms.

Otoshi et al. [40] proposed dynamic slice selection by learning to recognize the rough situation and the mapping between the current situation and the future slice. The Bayesian Attractor Model (BAM) was used to achieve consistent recognition, as well as the Dirichlet Process Mixture Model (DPMM) to achieve automatic attractor construction. Situations mapping was also automatically learned using feedback. The video streaming situation was used for the application of dynamic slice selection and the results show that the proposed method can maintain a high quality of video streaming while reducing slice changes. Extended BAM can be used to reduce the number of slice changes while reducing the quality degradation of video streaming. Additionally, the integration and deletion of attractors was used to maintain only the necessary number of attractors. Future work should include control lag as a parameter for the slice selection prediction mechanism.

## 4. Intelligent Edge Computing

Edge computing refers to a broad set of techniques designed to move computing and storage out of the remote central cloud (public or private) and closer to the source of data [41]. It exists to increase the computational capacity of the network. Supposedly, the next generation networks may support the connection density of more than 1 M devices/km^2^. The introduction of a large number of devices in the networks hinders the processing of a large amount of data within low latency [42].

New applications, such as tactile Internet services, may also require an extremely high data rate, lower latency, and extended reliability. To facilitate these types of services, edge computing is incorporated as a technique to process most of the data at the network edge, instead of transferring it to the cloud network that is far away.

Within the broad topic of edge computing, ETSI Multi-Access Edge Computing (MEC) is the widely accepted standard that must be met for a technology to be considered edge computing [43]. Its standards are guided by the following principles [29,44]:Edge computing must have a virtualization platform to be considered MEC (ETSI uses its NFV architecture in the standard);MEC can be deployed at radio nodes, aggregation points, or collocated with the Core network functions;APIs in a MEC environment must be simple, controllable, and, if possible, reusable for other tasks;Since the computing, storage, and network resources required by an MEC application may not match what is available in a node, an MEC network requires lifecycle management for the entire application system to handle these variables correctly;MEC systems must be able to relocate a high-end mobile application running on an external cloud to an MEC host and vice versa, meeting all the application requirements (ETSI admits that this principle needs further research).

Figure 2 shows the cutting-edge computing technology used to enable several next-generation applications. Differently from fog computing (a concept initiated by Cisco [45]), edge computing is associated with processing at the end of the device, rather than processing over the local network provided by fog computing. A fundamental difference between MEC and fog computing is that MEC works only in the autonomous mode, whereas fog computing has several interconnected layers and can interact with the distant cloud and the network edge [46,47].

The combination of Artificial Intelligence (AI) and edge computing was introduced to manage several emerging future communication problems. However, with the limited availability of resources and storage, it is very difficult to operate highly complex AI-based algorithms that require huge data collection at endpoints. Research is needed to design new and thin AI algorithms for edge nodes. In addition, the development of effective mobile resource scheduling and transfer techniques to improve system performance is also necessary [12].

Thus, by combining edge computing, AI, and NS we are able to meet the latency requirements of critical services, ensuring efficient network operation and service delivery with improved user experience. Furthermore, intelligent edge computing is important for slice selection as it is able to alleviate the network core functions, as the response occurs during applications in real-time, therefore reducing bandwidth usage with improved performance. In this sense, Section 4.1 presents the slices classification techniques and Section 4.2 discusses slice composition with ML.

### 4.1. Slices Classification Techniques

Network selection in a heterogeneous network environment represents a difficult problem, since there is no solution or technique accepted in this field yet, due to the number of variables and existing scenarios. This can be evidenced with the solutions that either consider or do not consider the process of inter-network mobility.

The implementation of new techniques for network selection thus becomes quite feasible, mainly due to the growing demand for its use in vehicular networks, patient monitoring, and smart cities, among other technologies and scenarios involving network convergence [14,15,48].

The most common methods reported in the literature for solving the problem of network selection include the use of fuzzy logic, Multiple Attribute Decision Making (MADM) or MCDM, Genetic Algorithms, and Artificial Neural Networks and ML. Among the MADM methods used, there exits the Analytic Hierarchy Process (AHP), Simple Additive Weighting (SAW), TOPSIS, Multiplicative Exponential Weight (MEW), Simple Multiattribute Rating Technique (SMART), VIKOR, COPRAS, Promethee II, and Grey Relation Analysis (GRA) [8,13,17,18,48,49,50,51].

Regarding the models that consider hybrid solutions, a feature that has achieved significant results concerns techniques that include fuzzy logic, MADM methods, and ML [6,16,17,34,36,52,53,54,55]. These models generally work in a similar way. After the process of data collection, according to the criteria outlined in the previous section, fuzzy logic processing occurs, followed by the classification method via a decision strategy. In this case, for each criterion, a certain weight is allocated to prioritize some services over others, guiding the choice of the new network in accordance with the application in use.

#### Multicriteria Methods

Multicriteria decision-making approaches consist of several techniques for solving problems. However, there are uncertainties, information conflicts, and disputes in the criteria that are required to evaluate the alternatives. These characteristics are inherent parts of the problem, and must therefore be mapped by the decision maker (person or system) to solve [56]. The objective is to evaluate a set of viable alternatives, through different criteria, considering for each a set of characteristics such as their respective weights; degrees of importance; or preferences of the decision maker [57].

The Promethee II method applies a concept called overclassification, which is a contribution proposed by the European School’s multi-attribute decision-making methods. Based on this concept, the alternatives are compared in pairs in order to infer whether a given alternative is as effective as the other. If alternative *a* is better than alternative *b* considering a given criterion, *a* is said to overclassify *b*. When all alternatives are compared, the final result is a classification (ranking) of all the alternatives in a given set, from best to worst. For this work, the second version of the method was chosen, as Promethee I includes situations where alternatives are incomparable. Promethee II inhibits judgments in which the alternatives are incomparable, always allowing for a complete classification of actions [13].

The TOPSIS method employs the principle of choosing an alternative that is closest to the positive ideal solution (best solution), and farthest from the negative ideal solution (worst solution). Thus, the method focuses on maximizing benefits and minimizing costs [35]. The method is widely used in the literature for different types of problems and has several extensions, in addition to hybrid models with other techniques [8].

The VIKOR method is based on the concept of compromise ranking, meaning it defines a measure of proximity with the ideal solution. In this context, the method uses a linear combination of Manhattan distance and Tchebychev distance metrics, where the former represents the maximum group utility, and the latter represents the minimum individual weight of the “opponent”. After obtaining the decision matrix and the weight vector, the method employs a set of mathematical operations, resulting in the ranking of the alternatives [56,58].

The COPRAS method seeks to assess the superiority of an alternative by implementing an evaluation ranking that considers the performance of the alternatives in relation to different criteria and their weights. The method can be used to maximize or minimize criteria, and it is used in several areas of knowledge [57].

The choice of these methods for validating the proposed framework is due to their popularity in the literature for different types of problems, applications, and areas of knowledge, in addition to their proven efficiency. The details of the methods and their mathematical operations are beyond the scope of this work but can be obtained from the previously mentioned references.

### 4.2. Slice Composition with Machine Learning

The allocation of resources for composing a slice must support the *slice* lifecycle management function [59], which, in turn, must meet the requirements specified in the SLA and the QoS and QoE rules in compliance with latency metrics, throughput and capacity of available resources [60]. Considering the characteristic of distributed computing [61], virtual systems and functions, the variable dynamics of the network, as well as the dynamicity and variance of the parameters over time, make the orchestration of resources for slice composition a complex and appropriate task for applying a solution based on AI [62]. However, a premise to apply AI techniques is the existence of a mathematical model, the knowledge of an expert, or a set of data. Given the inexistence of the first two options in this work, it was necessary to build or search for datasets that portray the slice efficiency metrics or even use data streaming for application to AI models.

Hence, monitoring a network’s incoming traffic or the historical behavior of slices in the form of raw data by streaming can represent the behavior of the network in terms of QoS and serve as a basis for obtaining optimal models. The research interest in resource allocation usually focuses on how to implement slice instances according to the description of resource requirements [63] and following the service agreement.

In this sense, a methodology for collecting and analyzing data from a network that is flexible enough to support the interactive and iterative model for producing ML Models is validated. Thus, a good strategy is to establish a data format to train various AI models and perform slice recognition and recommendations given the flow generated by the client.

The research interest in resource allocation focuses on how to implement virtual network instances under shared physical resources. End-to-end NS for 5G mobile networks using ML has intensively been studied [5,59,60,64,65,66,67]. One of the most important issues related to the use of slice by attribution performed from ML is to guarantee the generation of optimal models to avoid the sub-utilization of resources as well as the SLA breaking.

Establishing metrics and an optimal slice recovery mechanism for deterministic demands was the starting point in the work of Wen [68], in which he sought to build the ideal solution to be used as a reference in other algorithms. In stochastic demands, the option of optimization mechanisms that generally have slow convergence is initially adopted, with the use of ML as a future solution. Establishing a dataset that reflects an optimal solution for guiding ML is an appropriate scenario for building supervised learning models. However, the diversity of scenarios and demands of the clients of the networks make standardized definition a complex task.

Thus, exploring types of learning that are not dependent on datasets containing the optimal model classifier label is presented as a plausible and viable alternative. Hence, for example, unsupervised learning, recurrent, and convolutional neural network models are adopted in the construction of models for solutions related to NS. In the work by Toscano [69], neural networks of the Long-Short Term Memory (LSTM) type were used for slice provisioning in a simple mechanism with only four parameters contributing to the SLA, being (Rs, Ks, Ws and Ds), where Rs is the average ratio of resource utilization, where Rsϵ[0,1]. To ensure tolerance to the network operator, Ks measures the maximum standard deviation of resource sharing. In turn, Ws is a window of time during which the average is calculated and where the sharing of resources plus the deviation must be guaranteed. Finally, Ds specifies the lifetime duration of the slice. The training of the neural network was conducted by using data from 24 h of running a traffic simulation in the NS-3 Network Simulator [70]. When deploying the performance achieved in training the network, the author concluded with the need to generate more data to supply the LSTM network.

Cui [71] explores a variation of the LSTM architecture together with Convolutional Neural Networks (CNN) in a slice context approach for vehicular networks (V2X). The neural network training used a mobile network traffic dataset from the city of Milan, Italy [72], which contained data from the following three categories: SMS, telephone, and web browsing. For the simulation, each category considered a slice to be allocated through the model which, in turn, presented a satisfactory performance in establishing the connection between the customer and the slice that would serve it. The LSTM network is of the Encoder/Decoder type, and in the project in question, the encoder was used to predict the network traffic, while the decoder obtained the optimal slice. The entire datasetwas used for network training, with the input in the network being an image matrix. Despite the satisfactory result reported, the SLA was not presented, or the parameterization of the dataset attributes that was used. Therefore, replicating the procedure to validate the performance of the network would not be feasible.

## 5. Proposed Solution

The slice-selection task in the network runtime in a heterogeneous environment is a difficult problem since there is still no fully accepted solution or technique in this field. The reason for this is the number of variables and scenarios that exist, such as the case of solutions that consider or do not consider the inter-slice mobility process.

Thus, the implementation of new slice selection techniques becomes quite viable, including the demand for the growing use of vehicular networks, smart cities, robotics, agriculture 4.0, healthcare, remote surgery, Unmanned Aerial Vehicles (UAVs), IoT, and Internet of Vehicles, among other technologies and scenarios involving network convergence.

Overall, the literature suggests approaches that consider the following situation: Considering a given set of criteria or network parameters, verify at any given time and among the available slices, which one better fits the user needs, supporting the network exchange (handover process) for the mobile [8,36]. In this case, the process of choosing the slice is subject to certain criteria [5,17,18].

This work presents a novel approach that employs several techniques aiming at the integration and interoperability between RAN networks and the core of the proposed orchestration architecture, based on an efficient and robust NSSF that provides compatibility with the specification standards (e.g., 3GPP, ETSI NFVI, and 5G PPP).

### 5.1. NSSF DAF: Network Slice Selection Function Decision-Aid Framework

Figure 3 provides an overview of our proposed framework for NS selection. NSSF DAF consists of a solution in which a part is executed on user equipment (e.g., smartphones, vehicles, IoT brokers), running as a transparent service and another one runs at the edge of the network operator or service provider. The framework has several modules that can be configured according to the context of applications, geographic location, scenarios of mobility, strategies of slices selection, and others. For energy saving, the user equipment only signals its consumption profile or user application preference to the framework hosted at the edge of the architecture. Hence, no processing occurs in the mobile device or in the IoT broker.

According to Figure 3, the framework proposed was divided into the following three main blocks: *Collector*, *Processor*, and *Decision Maker* of NSs. Fundamentally, the criteria for network selection are closely related to the demands or applications in use. Thus, parameters that measure the application’s QoS as well as objective quality metrics for specific applications such as Quality of Video (QoV) and subjective metrics, such as indicators based on the user experience (QoE) and Mean Opinion Score (MOS), may be considered [15,73]. The *Collector* Module focuses on the assessment and dynamic mapping of the appropriate QoS requirements for each type of service, in addition to considering the signaling of the User Profile (e.g., V2X, Virtual Reality (VR), Augmented Reality (AR), Video on Demand (VoD), Video Stream), monetization and geographic location [15,16,54].

*NSSF DAF* is indifferent to the technique used to mark packages in gNodeB, that is, our architecture assumes that a software instance in Centralized Unit (CU) and Distributed Unit (DU) based on widely used solutions, such as Segment Router, Multi-Protocol Label Switching, and/or definitions of IPv6 classes of service have already marked (labeled) the packages [74]. Therefore, our framework only identifies and collects the QoS, QoE (from the return of the app), and MOS (the defuzzification process of the *Processor* module) parameters, processes them, and, from there, defines which NS best meets those requirements or in case of non-existence, signals to the Orchestration solution, the parameters for instancing the slice at run time (already in the cloud).

The Module *Processor* preferentially uses models that consider hybrid solutions, as mentioned in Section 4.1. In general, these models work similarly, that is, after the data collection process, the slices are classified and ranked, and finally, selection via the decision strategy is performed. In this case, each criterion receives a certain weight to prioritize some services over others, guiding the choice of the new slice according to the application in use. The definition of weights also takes into account the user profile and other aspects of the available slices. As described in [75], all computational methods are implemented using VNFs, running in an environment based on OpenStack [76] or Kubernetes [77].

For storing the data collected in the measurement processes, as well as the persistence of results related to the processing module, a NoSQL database is used. Data can be manipulated and analyzed by computational intelligence algorithms, using APIs (Application Programming Interface), and consuming data directly from the *Network Performance Analytics* module, based on the Hadoop [78] and Spark [79] ecosystems. Note that the framework supports any relational database.

The *Decision Maker* Module selects the slice that best fits the UE data stream and then aggregates the network traffic and redirects it to the slice that meets the required SLA configuration. All the operations between modules and blocks of the framework are performed via APIs, which provide external access by other entities through RSA key pairs (SSH keys).

Information about traffic conditions on slices, as well as extra options for configuring the structure, are available on a panel in the Graphical User Interface (GUI) or via Command Line Interface (CLI) when accessing the slice selection service. It is possible to combine more than one selection strategy or method, for example using genetic algorithms, fuzzy logic or multi-criteria decision methods. It is also possible to define which QoS parameters should be considered in the selection process.

### 5.2. Structure Specification

The NSSF DAF is structured around three large blocks or main modules, as shown in Figure 4.

The *Collector* module presents a set of components responsible for data acquisition and pre-processing. The characteristics and functionality of these components are detailed below:*Data Collection*: The component presents a set of methods responsible for obtaining the QoS metrics (e.g., Latency, Jitter, Packet Loss, Reliability, Bandwidth), QoE, and user consumption profile (e.g., VoD 4K, AR/VR, V2X). This component uses passive measurement techniques, using packet analyzers and protocols (sniffers). It can be configured to use active measurements through ICMP requests, collecting information such as Round-Trip Time (RTT), and other metrics. The module offers data consumption routines through publish/subscribe solutions.*Data Cleaning*: This component is responsible for data cleaning, the fixing of reading errors, and dataset structuring.*Validation*: The component responsible for verifying that the set of metrics received satisfies what is expected from a dataset.*Transforming*: The Transforming component has the following three subcomponents: Scaling, Typing, and Encoding. The Scaling subcomponent performs the normalization process with libraries and methods, which facilitates the application of machine learning techniques. The Typing subcomponent makes the necessary data typing conversions (e.g., float to integer, integer to float, string to float). Finally, the Encoding subcomponent performs the necessary transformations in categorical metrics by converting them to numeric classes, in order to include them in the machine learning models.

The *Processor* module presents a set of components prepared to use several optimization techniques, computational intelligence, and stochastic and multi-criteria decision approaches. The characteristics and functionality of these components are detailed as follows:Fuzzy Logic: The component supports the use of fuzzy logic, its membership functions, fuzzification, and defuzzification methods. The *Collector* module output allows for component integration transparently.Genetic Algorithms: The component allows the use of genetic algorithms. The *Collector* module organizes the data, thereby facilitating integration, the definition of generations, and an initial population. It also facilitates the application of multi-objective formulations directly to the dataset.Machine Learning: The component supports the use of machine learning techniques. Data organization performed by the *Collector* module facilitates the use of supervised models. It also allows reading non-relational databases, batch files, or data streaming, which allows using unsupervised models.MCDM Methods: The component allows the use of several multi-criteria decision methods. The structuring of datasets facilitates the assembly of decision matrices. The weights of the criteria can be defined directly by the network operator or service provider, and can also be obtained using hybrid approaches with machine learning techniques. Weights can even be mapped directly from the end user’s consumption profile. These integrations take place through APIs.

The *Decision Maker* module presents the necessary components needed to integrate the techniques used in the *Processor* module, with Cloud Computing tools and solutions, especially the network orchestrators. The characteristics and functionality of these components are detailed below:*Generator SLA*: The component is responsible for generating the SLA that will be used in the creation of on-demand slices (horizontal model), or for forwarding data flows to the already available and pre-configured slices (vertical model). The SLA is assembled from the output of the *Processor* module and fulfils the requirements of the application in use or the user’s consumption profile.*Export Data Model*: This component uses multiple data models which facilitates integration with third party tools, especially network orchestrators. The transport of data models takes place via the consumption of APIs.

### 5.3. Integration Overview

Figure 5 presents an overview of the integration of the proposed solution in the context of 5G networks. NSSF DAF was initially designed to function as an MEC service, being one of the main contributions of this work. However, the objective was to evaluate the implementation in a local datacenter close to the gNodeB in the RAN, or in the 5GC. It seeks to verify whether there is a significant difference in the performance of slice selection at different points of the network slicing architecture. Its integration with the data plane (UPF—User Plane Function) is essential for traffic recognition and the differentiation of service classes, which allows for its operation.

Among the main functional and non-functional requirements (*R*) of the NSSF DAF we identified the following:R1:Deployable on servers with different virtualization solutions (NFVI–NFV Infrastructure);R2:Facilitate deployment in production networks, in a transparent way;R3:Facilitate the integration of later modules, adding new features, through the available documentation;R4:Compliance with the specifications and models of the standards bodies (e.g., 3GPP and ETSI);R5:Act independently of specific hardware manufacturers and models;R6:Use open-source solutions;R7:Use DevOps—Development and Operations—philosophy, practices, and tools;R8:Make the mobile application available for download from repositories.

The client application is cross-platform and can be embedded in several UEs, such as UAVs, vehicles, smartphones, robots, or even in IoT brokers in smart homes. The application uses a set of APIs, providing the necessary signaling for the NSSF DAF. The UE sends messages with its location (GPS coordinates), checking if there is an occurrence of NSSF DAF on the Edge or RAN to which it is linked. From this signaling and the profile of the application used, the NSSF DAF selects the best slice available, and returns with the best option for the UE, as detailed in the modules included in Section 5.2.

The functional and non-functional requirements (RA—Requirement Application) of the client application are detailed below.

RA1:  Be able to search for the available slices (RAN and Edge);RA2:  Direct a request for the best slice option to NSSF DAF, through HTTP requests and API consumption;RA3:  Execute the handover to the best available slice;RA4:  Facilitate the setting of the user’s profile (preferences), which can be as follows: VoD 4K/8K, AR/VR, V2X, among others;RA5:  Run in automatic (service) or manual mode, according to the UE.

Regarding integration with cloud computing tools, the NSSF DAF operates with standardized data models, allowing for the required integration. The specification of the functionalities of all modules is detailed in Section 5.2.

### 5.4. System Scalability

The proposed framework is scalable and can be used with new criteria for slice selection, as well as different access network technologies. The interaction between NSSF DAF instances, even in different administrative domains, provides a network of information about the quality of available slices in certain regions (e.g., neighborhoods, universities, cities).

*NSSF DAF* also allows for the addition of new modules, for example, a broker service for IoT devices that support Message Queuing Telemetry Transport (MQTT) or other publish/subscribe protocols. It can also facilitate the attachment of a Mobility Management (MM) module, selection of Access Points (APs) for RAN based on Wi-Fi 6.

Framework deployment can be performed using DevOps tools, and it can still be deployed in containers or virtual machines on a public cloud or local datacenter. This feature can be used on private 5G networks.

### 5.5. NSSF DAF Restrictions

Although the NSSF DAF framework solves the slice selection problem, it does not address issues related to MM. Thus, the mobile device must also use a management protocol that allows for handover between slices, keeping the continuity of the service active, especially in instances sensitive to latency and packet loss, such as traditional real-time applications.

In [80] the main challenges involving MM are presented, as well as potential solutions. Among the main issues, we can mention handover signaling, network slicing, integration with other frameworks, and frequent handovers, among others. Potential solutions include the use of deep learning, the creation of mechanisms on demand MM, Software-Defined Networking (SDN)- NFV integration with legacy methods (e.g., IEEE 802.21, PMIPv6, LTE MM) and the use of edge computing solutions.

These demands are outside the scope of this work, but the framework proposed here is in line with these demands, and can be easily extended to cover these issues. This is mainly possible due to its deployment through the edge computing model, SDN and NFV, in addition to slice selection strategies, Radio Access Technologies (RAT), the network metrics acquisition approach and slice evaluation with traffic flow during operation.

## 6. Experimental Evaluation

In order to validate the proposed solution, we conducted experiments integrating an approach that uses ML and decision-making methods to support NSSF DAF operations in multi-domain slicing environments. The methods proposed in this approach focus on the evaluation and dynamic mapping of adequate QoS requirements by the analysis of network traffic coming from the UEs and then composing a template to create slices in various formats (e.g, YAML, NETCONF, and JSON), allowing for their integration with 5G Core network functions, and therefore the use of coding, automation, and network service orchestration tools, such as Ansible, Chef, and Puppet. The NSSF DAF framework also facilitates integration with ONAP Istanbul/Honolulu and OSM 11 orchestrators.

### 6.1. NSSF DAF: Network Slice Selection Strategy

The NSSF DAF framework processing module was developed by implementing a set of MCDM methods in Python language version 3.8, with the VIKOR, COPRAS, TOPSIS, and Promethee II methods. The initial choice of these methods was made due to their robustness, allowing for the validation of the proposed solution modules. However, other strategies to process data from the *Collector* module can be applied.

The Algorithms 1 and 2 constituting the most important part (core) of the proposed framework. Algorithm 2 is responsible for slice selection. Initially, we used the function ParserStream2DataFrame that applies Algorithm 1; then, the dataframe to be used in the preference calculations was obtained. The applyMethods function applies the defined decision-making methods.

The *applyMethods* function (see Algorithm 2) receives the original dataframe for parameters of the *Collector* module, which can be in the form of a data stream or batch file. The weight vector for each of the criteria used by MCDM methods and the characterization of the criterion type, that is, if the QoS parameter is of the type “the bigger the better”, which is represented in the array as 1 or “the smaller the better”, which is defined in the array as −1, in addition to the extra parameters of the MCDM methods were applied.
**Algorithm 1:** Parser Stream to dataframe
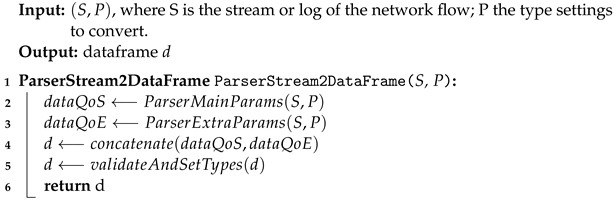

**Algorithm 2:** NSSF Decision-Aid Framework.
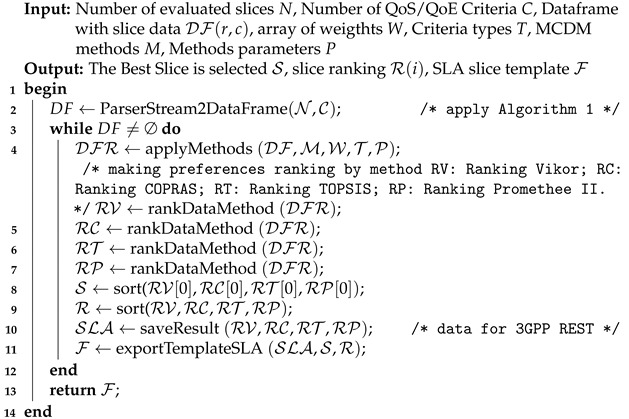


Furthermore, the algorithm integrates the extra parameters of the implemented decision methods. For example, for the Promethee II method, the usual criterion curve was considered because the dynamics of the parameters and the traffic conditions of the networks change dramatically in short periods. For the other methods, default configuration was used. The details of the VIKOR, COPRAS, TOPSIS, and Promethee II methods are beyond the scope of this work (more details can be found in [13,56]).

The *rankDataMethod* function defines the ranking of the best slices for each of the methods evaluated. This method implements the process described in Section 5.2 of the *Processor* module in the multi-criteria decision strategy (MCDM Methods). The other functions (*saveResult* and *exportTemplateSLA*) presented in the algorithm are responsible for handling the resulting data and configuring the SLA template in a 3GPP REST format. These functions facilitate integration with 5G CORE and, therefore, with orchestration tools. All computational methods are implemented using VNFs, running in an environment based on OpenStack or Kubernetes [75].

### 6.2. Environment Description

Given the need for a network traffic dataset to explore, validate, and develop ML models, this work followed the standard behavior presented in several articles by building the scenario and producing a dataset based on a simulation. This behavior was understandable due to the wide and diverse characteristics of computer network scenarios and needs.

The test environment was implemented using the network software emulator Graphical Network Simulator 3 (GNS3) to provide the infrastructure for the virtualized network functions (NFV Infrastructure). The scenario was composed of 05 UEs transmitting at different traffic rates, as specified in Table 2, for the set of collections performed. In addition, implementation in the gNodeB was performed through a software module (CUPS—Control User Plane Split) running on an Ubuntu Linux 16.04 LTS VM (Maintained by Canonical based in London, UK) with Kubernetes, based on the OpenAirInterface (openair-k8s) project [81].

Regarding the network edge (Edge Computing), a docker container was used with the deployment of the SDN controller OpenDayLight, to implement the UPF, responsible for the separation and identification of data flows. The Network Data Analytics Function (NWDAF) was implemented in an Ubuntu Linux 16.04 LTS VM with the Anaconda Python 3 framework, to implement the analysis functions and the data mining pipeline, using the Jupyter notebook tool and the Numpy, Pandas and Sklearn libraries.

Although the scenario presented the core of the 5G architecture (5G Core—5GC), none of the Network Functions (NFs) relevant to the core were implemented, such as Network Exposure Function (NEF), Network Repository Function (NRF), Policy Control Function (PCF), Unified Data Management (UDM), Authentication Server Function (AUSF), Access and Mobility Management Function (AMF), Session Management Function (SMF) and Application Function (AF), as shown in Figure 6. These network functions will be implemented through ETSI NFV OSM, which will use the SLA template generated by the NSSF DAF framework.

However, three vertical slices were considered, as detailed below, delivered at the network edge by three different 5G providers, according to the 3GPP TS 22.186 V16.2.0 and 3GPP TS 22.261 V16.14.0 specifications [82,83]. The composition of each of the slices is given by the junction of Virtual Networks (VNs)’ N1 to N6, under the domain of three different Internet Service Providers—ISPs (X, Y, Z) according to Table 3. Regarding the representation of services, an Iperf server was used for different traffic transmission rates of the Constant Bit Rate (CBR) type.

As a result, for each of the 11 collections performed in the simulation process, a trace file was obtained for each UE containing a dataset of 33 iterations. Iteration data comprises: the amount of transferred bytes, bandwidth used, transport protocol, latency, jitter and packet loss. Each collection was carried out for a period of 5 min. These raw log data are thus the starting point for carrying out the data analysis process.

### 6.3. Data Analytics

In the traditional data analysis process, falling in love with the dataset is a part of the first step. Hence, to know and to prepare the data for the mining process, we conducted the tasks of data pre-processing as follows:Simulation data collection;Data selection;Data purification;Dataset construction.

Algorithm 1 details the process discussed above. The experiment considered six attributes organized in the datasets. However, the network provider can add metrics to suit different scenarios as long as it can collect data from the network. To validate this principle, two other criteria were added to the dataset, namely reliability and distance. These, in turn, were randomly generated, respecting a pre-defined interval within what normally exists in real scenarios according to the 3GPP TS 22.186 V16.2.0 and 3GPP TS 22.261 V16.14.0 specifications [82,83]. Thus, the default dataset now has the following attributes:Latency: End-to-end delay;Jitter: Delay variation;Loss: Packets loss;Bandwidth: Number of bits per second (Mbps);Transfer: Amount of data transferred (sent);UE: User Equipment, not used in the model building process;Experiment: experiment id, not used in the mining process;Distance: Shortest path between UE and Edge;Reliability: Capacity network is functional without interruption.

In the pre-processing step, the data are analyzed regarding the type and interval; treatments such as normalization are commonly applied to the records in this phase. The occurrence of treatment and normalization actions applied to the records is common. The data can thus come to serve and adapt to the input pattern of the data mining algorithm used future steps.

In order to recognize the data, for illustration purposes, Figure 7 presents the network metrics. Note that data were normalized using the *MinMaxScaler* method from the Sklearn library [84].

The mining process used the K-Means classification or grouping algorithm that uses numerical data. The "protocol" metric is not used because of the distance-based K-Means characteristics. The learning process now considers only the attributes relevant to the experiment: [“Latency”, “Jitter”, “Loss”, “Bandwidth”, “Transfer”, “Distance”, “Reliability”].

Clustering performs data analysis to recognize the group behavior of the data so that an element is characterized in one group and differs from the other groups. The K-Means k indicates how many groups are separable in the dataset. One technique to find the correct value of k for a given set is to apply the Elbow Plot Method by calculating the sum of the Root Mean Square Error (SSE). The Elbow point occurs when the SSE starts to decrease linearly. Thus, in the dataset used, the appropriate k is 3, as shown in Figure 8, with the SSE for k from 1 to 8.

Finally, assuming a k equal to 3, the learning and training process occurs with the separation of elements into three groups that characterize each slice. The next step to data mining is analysis, which seeks to understand the results. To clarify this step, the illustration in Figure 9, Figure 10 and Figure 11 presents the unique characteristics of each cluster that, in a way, reflect the characteristics of each slice used in the GNS3 simulation environment. Once the model is trained, the system can receive any data stream within the average training interval to provide the suitable *slice* prediction for the allocation.

In the analysis stage of the results produced by the mining process, the ability to generate illustrations is explored, always with the aim of validating the acquired knowledge. Such knowledge is useful, new, and applicable to the context of the problem.

### 6.4. Definition of Algorithm Parameters

After performing the collection and analysis of the network parameters, the application runs for the slice selection. This flow can be seen in the architecture of the proposed framework, illustrated in Figure 3.

In summary, the configuration of the parameters defined in Algorithm 2 for carrying out the tests is presented in Table 4. The weights were defined in the following order of parameters: [“Latency”, “Jitter”, “Loss”, “Bandwidth”, “Transfer”, “Distance”, “Reliability”]. The configuration of the weights followed the results from the K-means algorithm, which showed the most relevant features for each slice (obtained clusters), as illustrated in Figure 9, Figure 10 and Figure 11.

Thus, for Test 1, the Latency and Reliability criteria were considered with higher weights. Test 2 considered the Latency and Loss criteria, and Test 3 considered the Bandwidth and Transfer criteria. Finally, Test 4 used a fair distribution of weights between the criteria evaluated and Test 5 used random distribution. Each test is formed by 250 iterations in which the values of the criteria evaluated change according to the data flows from the UEs.

### 6.5. Results and Discussion

To understand the results measured by each approach, we observed the behavior of the criteria in the flows of networks. To this end, we used the boxplot statistical resource to evaluate the main QoS criteria, such as the symmetry of the data, its dispersion, and the possible presence of outliers. In addition, we used the histograms feature to verify the distribution of the other criteria considered.

Note that Figure 7 presents several outliers for the jitter and loss criteria. This is characterized by their stochastic, dynamic, and nonlinear behavior and the variation in traffic rates considered in Table 2.

After this preliminary analysis of the behavior of the criteria adopted as input parameters of the implemented framework, we evaluated the four approaches concerning their capability of classifying the best slice, considering the weights or preferences defined a priori by the *Decision Maker*, and also the traffic requirements of the UEs.

Table 5, Table 6, Table 7 and Table 8 detail the results for each method used in the NSSF service.

Considering the results generated using the VIKOR method, as detailed in Table 5 and illustrated in Figure 12, Slice 1 (Remote Driving) was selected as the best in four of the five tests performed. For the variation of weights of Test 3, Slice 1 was selected in 78.4% of the iterations and in Test 4 it was selected 93.2% of the time. This behavior emphasizes the sensitivity of the Latency and Reliability criteria since these two parameters are critical for the Remote Driving service.

The results from the COPRAS method are described in Table 6 and shown in Figure 13. Slice 3 (ITS) was selected as the best in four of the five tests performed, followed by Slice 1 (Remote Driving) in Test 5. Note that for Test 3, Slice 3 was chosen in 98.4% of the iterations. In the VIKOR method, Slice 1 had a higher selection ratio, followed by Slice 2. This behavior also occurs in Test 3 from the TOPSIS and Promethee II method. Slice 3 was selected at rates of 99.2% and 99.6%, respectively. This characteristic demonstrates the greater flexibility in the classification of alternatives for these methods as a function of the weights adopted.

Regarding the results from the TOPSIS method (Table 7), which are illustrated in Figure 14, we observed a closer distribution between the percentages of slices choice, except for Tests 3 and 4, in which there was a majority preference for Slice 3, of 99.2% and 76.8%, respectively. The same behavior also occurs in the COPRAS and VIKOR methods.

For the Promethee II method, detailed in Table 8 and illustrated in Figure 15, there is a homogeneity in the selection of slices for Tests 1 and 2, a practically unrestricted preference for Slice 3 in Test 3, with the same slice chosen in 99.6% of the time. In general, throughout the iterations of each of the tests, Slices 2 and 3 presented better resource conditions to receive the UE flows.

To verify the performance of the MCDM methods in the slice selection problem, some other graphs were generated, aiming to more succinctly illustrate the performance of each approach regarding the tests executed (please refer to Figure 16, Figure 17 and Figure 18).

Once the data results from each method were obtained, a descriptive analysis checked for significant differences in the performance of the methods for the slices evaluated in the set of tests. The experiment consists of comparative analysis, using Tukey’s test of multiple comparisons of means, from the analysis of variance. We also applied the Shapiro–Wilks normality, Durbin–Watson independence, and Fligner–Killeen homoscedasticity tests [85].

From the analysis of the test results, the Tukey test revealed that the selection means between the methods do not differ significantly, as discussed above. Another point of interest refers to the results from the Shapiro–Wilks normalization test that describes a normal distribution of the samples. For all the tests performed, there was no violation for Shapiro–Wilk. From the Durbin–Watson test, we can state with 95% confidence that the residuals are independent. The Fligner–Killeen test shows the samples have homoscedasticity of variances.

## 7. Conclusions

This work presented a comparative study between some slice selection strategies. We used a test scenario including 3 vertical slices in a multi-domain scenario, aiming at three common services in the context of 5G networks. In addition, we used a strategy from the K-means method to group the UE data flows, so as to characterize the available slices and, subsequently, define the weights that would be used in the MCDM methods, whose purpose consisted of defining and selecting the best slice available to receive the network flow during operating time.

Initially, it was necessary to study the slice selection models available in the literature, verifying the techniques that aim to solve the problem, so as to support the choice and option for the methods covered herein that constitute the core of the NSSF DAF framework.

The MCDM methods proved to be favorable, due to their simplicity and efficiency when compared to other strategies; they did not require a history of virtual network operations and slices, which can be very useful for newly-created slices. To define the weights and priorities between the criteria, the K-means clustering algorithm was considered, providing the opportunity for further studies and to adopt hybrid methods, considering the possibility of exploring reinforcement learning/genetics algorithms, fuzzy systems, or even networks recurrent along with the MCDM methods in future works.

Considering the restrictions of the scenario and from the analysis of the results, it was verified that the proposed solution proved to be promising, making it necessary to carry out new tests considering other scenarios, to propose improvements to the algorithms developed, as well as to evaluate other techniques, strategies, and methods for the core of the proposed framework.

We also observed that there is no significant difference between the evaluated methods. It is thus possible to evaluate the adoption of data grouping models, aiming to select the most suitable slice for the user considering the traffic generated. Hence, we proposed an optimized E2E approach for the traffic management of access networks to the 5G Core.

Finally, we concluded that selecting slices is still an open problem, which invites the study and application of new techniques, approaches, and solutions. Our proposed framework provides compatibility with current specification standards, thus constituting a promising solution for 5G networks in the context of NS.

Our proposal can also be applied in several application niches, such as when integrating large amounts of data from devices linked to the IoT context to cloud services, as well as to integrate applications that have multimedia requirements, a high density of video, and audio traffic.

### Future Work

As future work, we plan to develop non-linear models for the proposition dynamic and time-varying systems. This will allow the solution developed in our work to be implemented in real operation networks without change in their cores.

We intend to refine the proposed solution by evaluating other selection-based strategies for slices that constitute the core of the framework, such as genetic algorithms, deep learning, and utility function, among others.

We also intend to implement our refined solution for a mobile device, e.g., Vehicle, UAV, Robot. Our main goal will be to assess the solution at various points in the 5G network architecture, and thus statistically verify if there is a significant performance difference in the NSSF according to the implementation model.

Furthermore, we intend to perform experiments evaluating the integration between market orchestration tools, such as ONAP, EMCO, and OpenNESS. The integration model should be flexible, facilitating deployment anywhere in the network, without the need for changes to equipment and VNFs currently operating.

## Figures and Tables

**Figure 1 sensors-22-06066-f001:**
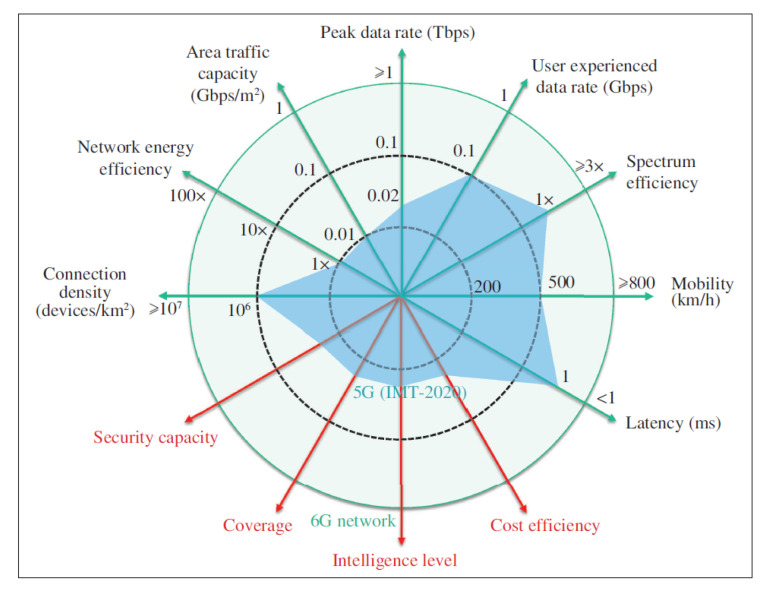
5G vs. 6G Network Requirements. Based on [12].

**Figure 2 sensors-22-06066-f002:**
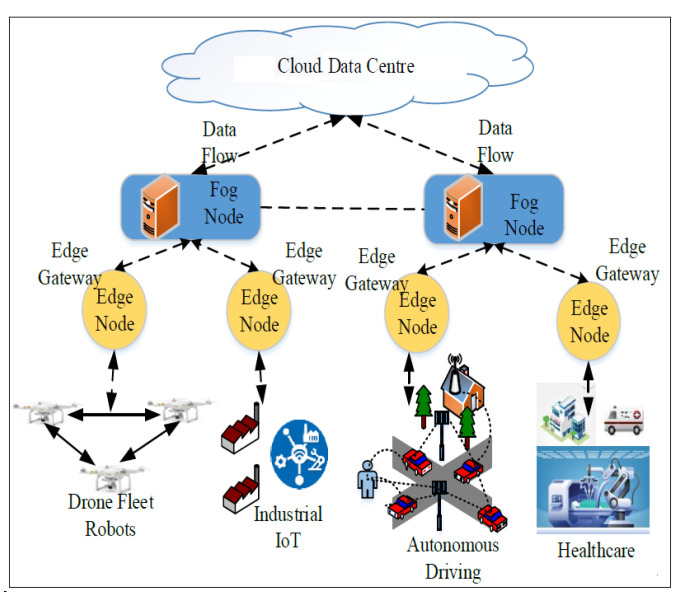
Edge computing and Fog computing interaction with some applications. Based on [46].

**Figure 3 sensors-22-06066-f003:**
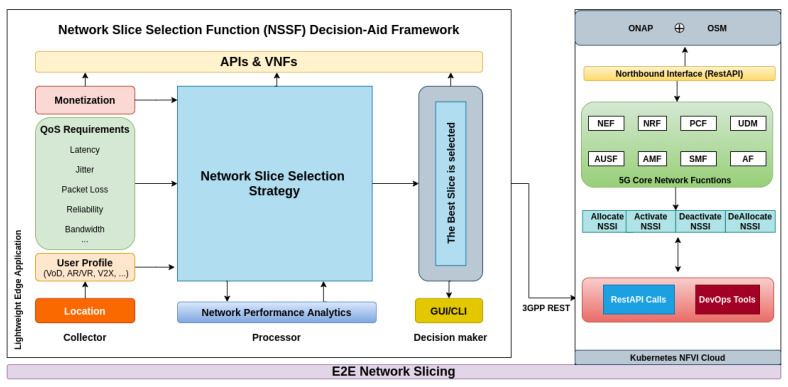
Proposed NSSF DAF: Network Slice Selection Function Decision-Aid Framework.

**Figure 4 sensors-22-06066-f004:**
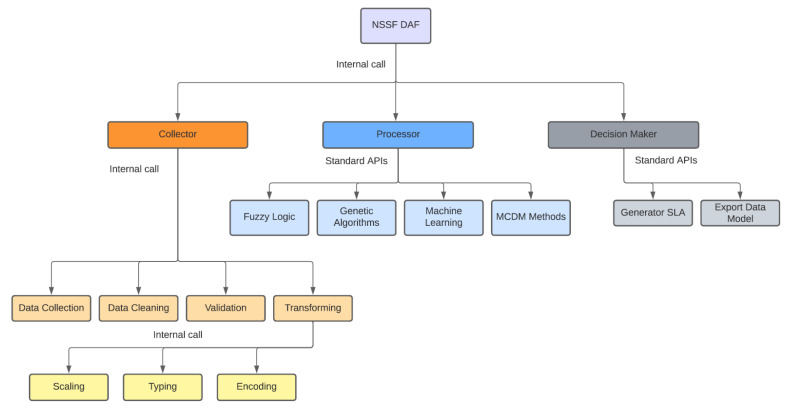
NSSF DAF Structure specification.

**Figure 5 sensors-22-06066-f005:**
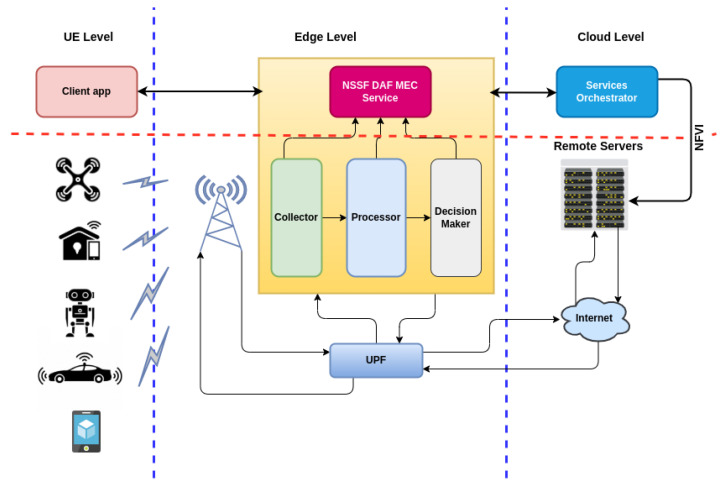
NSSF DAF Integration overview.

**Figure 6 sensors-22-06066-f006:**
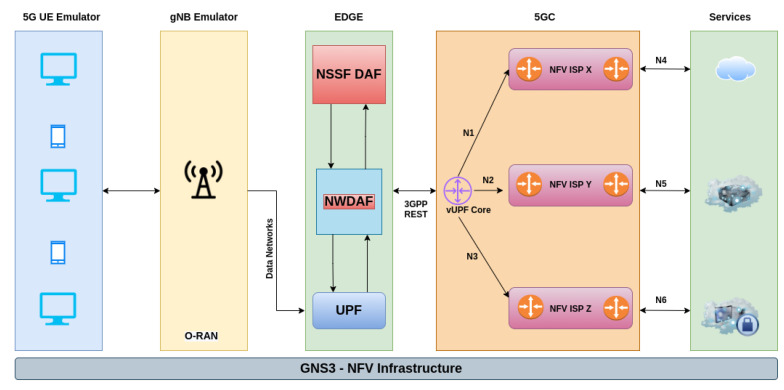
Illustration of the simulation environment architecture for producing network traffic.

**Figure 7 sensors-22-06066-f007:**
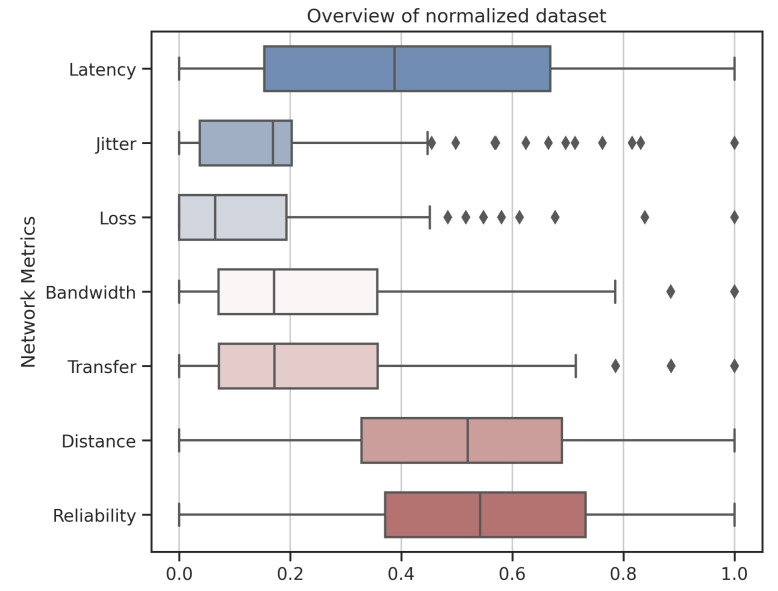
Overview of normalized dataset.

**Figure 8 sensors-22-06066-f008:**
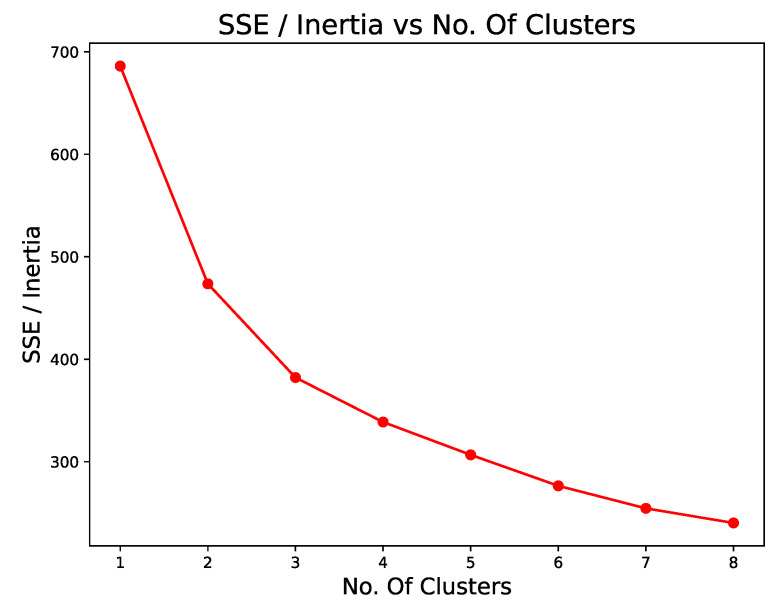
Downward Mean Square Error for K = 1 to K = 8.

**Figure 9 sensors-22-06066-f009:**
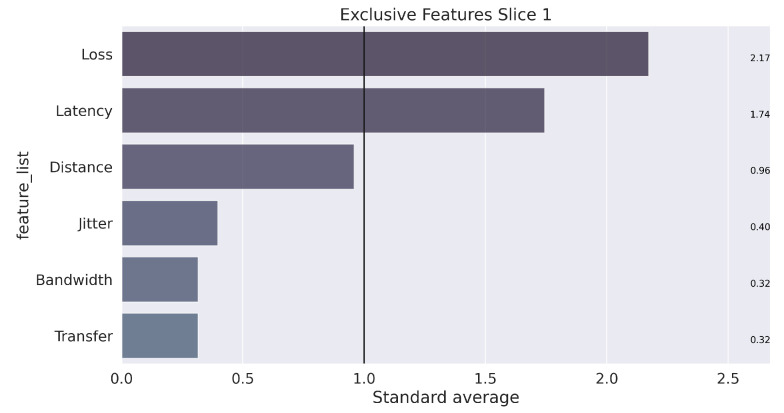
Unique characterization of the groups, found in the process for Slice 1.

**Figure 10 sensors-22-06066-f010:**
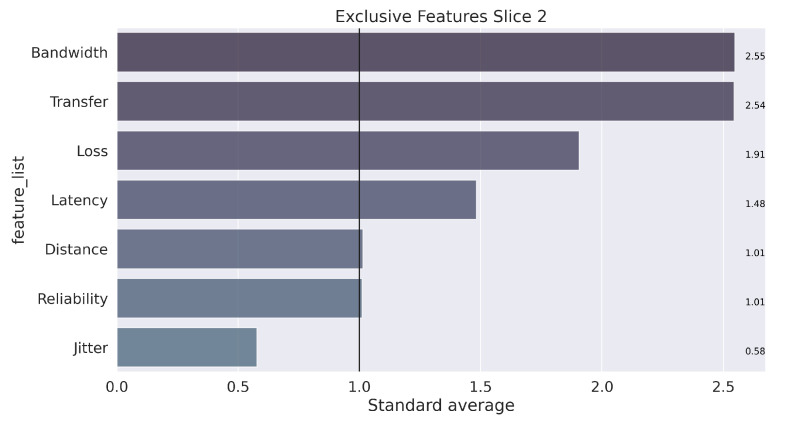
Unique characterization of the groups, found in the process for Slice 2.

**Figure 11 sensors-22-06066-f011:**
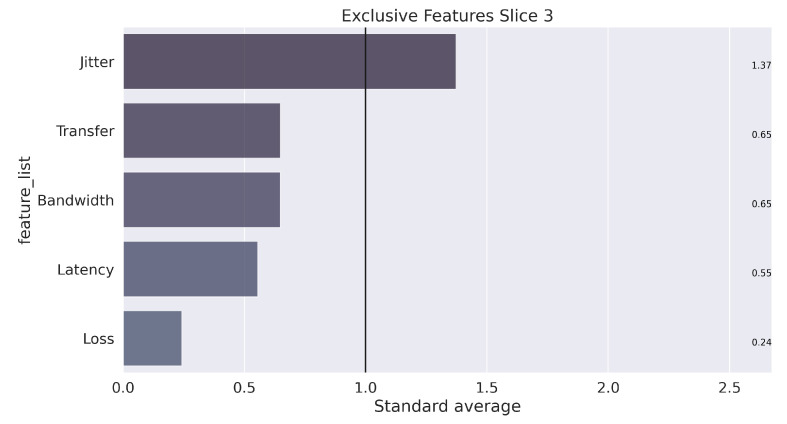
Unique characterization of the groups, found in the process for Slice 3.

**Figure 12 sensors-22-06066-f012:**
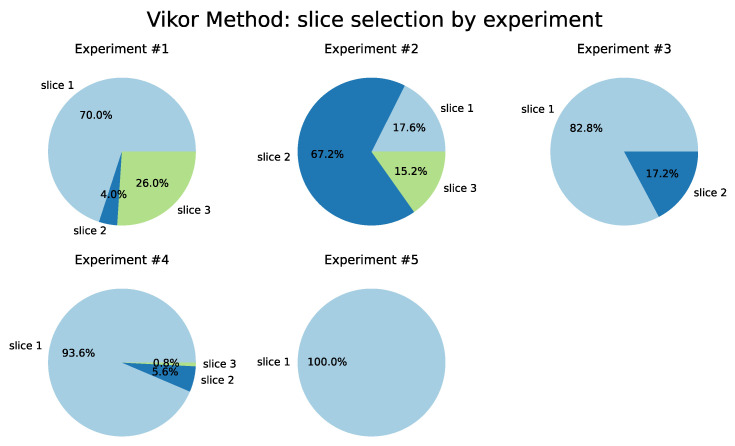
VIKOR Method.

**Figure 13 sensors-22-06066-f013:**
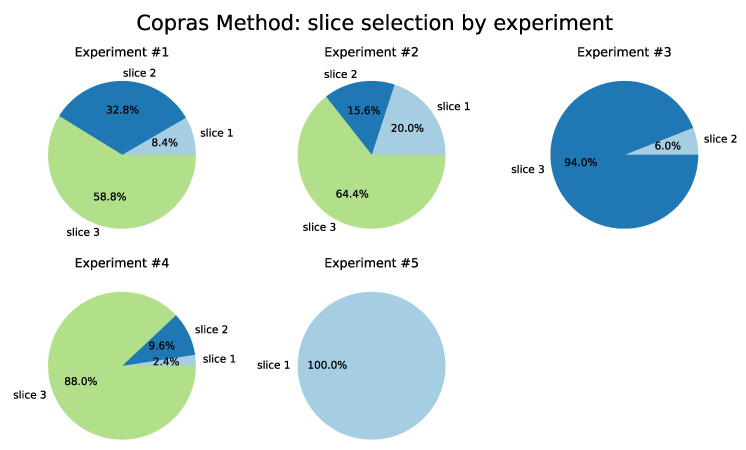
COPRAS Method.

**Figure 14 sensors-22-06066-f014:**
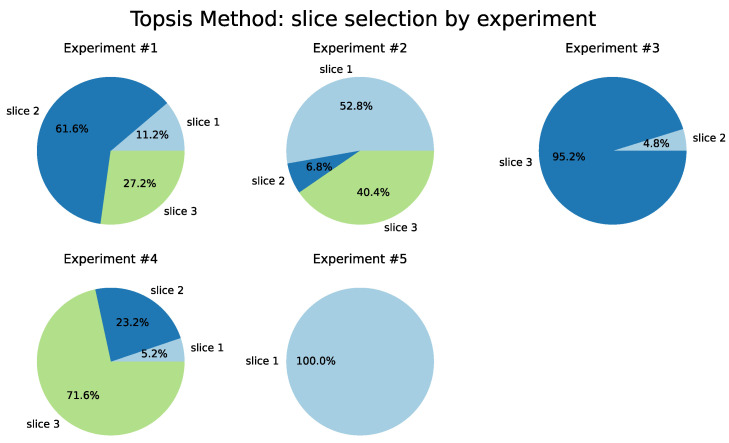
TOPSIS Method.

**Figure 15 sensors-22-06066-f015:**
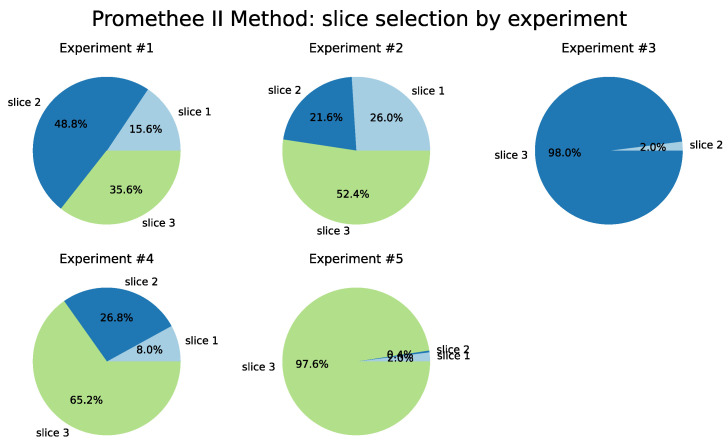
PROMETHEE II Method.

**Figure 16 sensors-22-06066-f016:**
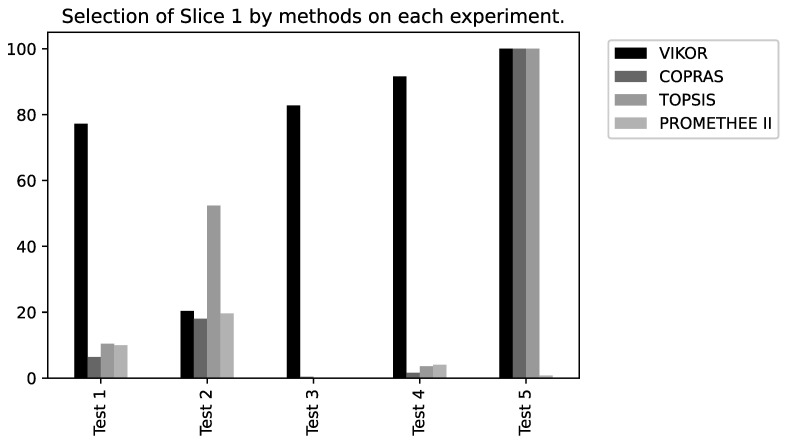
Method percentage results for Slice 1.

**Figure 17 sensors-22-06066-f017:**
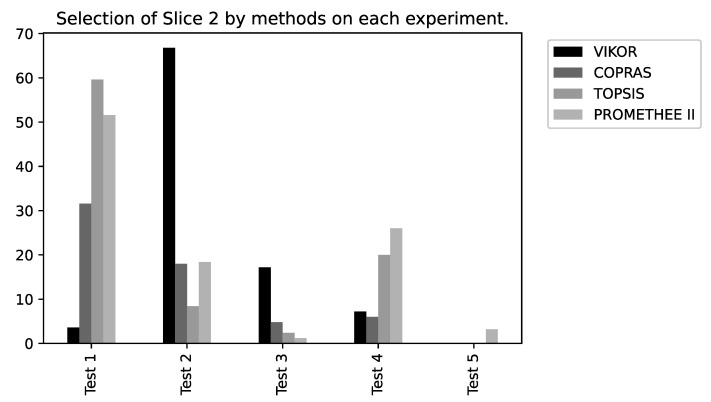
Method percentage results for Slice 2.

**Figure 18 sensors-22-06066-f018:**
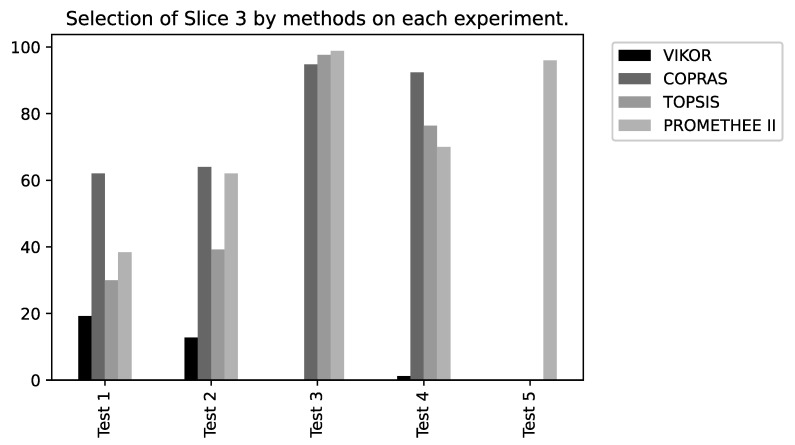
Method percentage results for Slice 3.

**Table 1 sensors-22-06066-t001:** Summary of Related Work.

Work	Contribution	Limitations	Future Works
Rivera et al. (2019) [34]	Traffic control rules can be deployed using minimal resources without influencing the efficiency of the Data Plane slices. A management system is necessary for the provision, updating, and control of the physical layer of VNFs that comprise the slices.	Lack of real world implementation	The inclusion of a monitoring agent that can oversee the status of SPGW-U traffic in real-time
Bojkovic, Bakmaz and Bakmaz (2019) [8]	SD-TOPSIS presents better performance in rank reversibility, but E-TOPSIS has significantly lower complexity operations. The E-TOPSIS method is suggested, especially when dealing with the fine granularity of slices. TOPSIS is considered a good decision-making tool, considering its algorithmic logic and mathematical form. Yet, it fails to provide consistent results due to the rank reversal phenomenon.	Lack of real world implementation	To analyze the influence of various alternative normalization techniques and ranking methods on NSSF performance
Bakmaz, Bojkovic and Bakmaz (2020) [35]	Alternative techniques, such as linear normalization (MAX-MIN), the weighting of variance, and binary classification alternatives, can reduce both the classification reversibility and computational complexity. This justifies the need to consider Multiple-Criteria Decision-Making (MCDM) methods as a potential solution to the network slice selection problem.Lack of real world implementation To analyze the performance of other MCDM algorithms in terms of ranking reversibility.	Lack of real world implementation	To analyze the performance of other MCDM algorithms in terms of ranking reversibility
Shurman, Rawashdeh and Jaradat (2020) [36]	A mechanism that enables user equipment to run multiple sessions on different network servers at the same time to utilize the advantages of their services.	Only a temporary session is allowed	To give users a PURE connection to the networks, with user registration and full capabilities
Dimolitsas (2020) [37]	A multicriteria decision framework for the optimal selection of Edge Points of Presence (EPoPs) to deploy a network slice. Results indicate the relevance of the proposed two-stage method in meeting the user’s hard and soft requirements, allowing communication between slices and optimal resource allocation from the providers.	High cost of deployment	Implement a distributed PoP selection mechanism, improving service discovery and cross-slice communication
Zhao et al. (2020) [38]	A Genetic Algorithm that can achieve satisfactory results in the maximization of user’s Satisfaction Degree (SD) in the E2E network slicing problem. This method obtained better access and transmission performance when compared to traditional selection methods based on the Received Signal Strength (RSS) or greedy algorithms.	Lack of real world implementation	It is reasonable to propose a real world experiment to validate the GA algorithm simulation results
Otoshi et al. (2021) [40]	A dynamic slice selection technique that learns to recognize the rough situation and the mapping between current situation and the future slice. The Bayesian Attractor Model (BAM) is used to achieve consistent recognition, as well as the Dirichlet Process Mixture Model (DPMM) to achieve automatic attractor construction. Situations mapping is also automatically learned by using feedback.	Problems such as the bit rate drop should be predicted in advance and the slices should be switched in advance	To incorporate the control lag in the slice selection prediction mechanism
Silva et al. (2022) [39]	The use of hybrid machine learning algorithms and MCDM methods as a solution for the 5G network slice selection in IoT scenarios. The proposed solution proved to be efficient and the adopted MCDM methods show a similar performance.	Restrictions of the test environment	New experiments considering different scenarios and further development of the proposed algorithms

**Table 2 sensors-22-06066-t002:** Specification of injected traffic during 5 min for each collection.

Collection	UE 1 (Mbps)	UE 2 (Mbps)	UE 3 (Mbps)	UE 4 (Mbps)	UE 5 (Mbps)
01	20	47	07	09	06
02	22	45	06	12	08
03	24	49	09	10	06
04	32	55	14	18	12
05	25	50	10	05	02
06	45	70	30	05	03
07	12	25	05	20	30
08	20	22	24	02	07
09	08	28	05	10	03
10	32	62	22	15	08
11	10	15	05	02	15

**Table 3 sensors-22-06066-t003:** Technical specification for slices composition. Based on [82,83].

Slice ^1^	Type	E2E Latency (ms)	Reliability (%)	Data Rate (Mbps)
1 (VN N1 → N4)	Remote	5 (maximum)	99.999 (minimum)	DL: 1 (minimum)
	Driving			UL: 25 (minimum)
2 (VN N2 → N5)	Rural	Not specific	Higher than 80%	DL: 50
	Macro			UL: 25
3 (VN N3 → N6)	Wireless Road-Side	30 (max.)	99.999	10
	Infrastructure			
	Backhaual (ITS)			

^1^ VN: Virtual Network; DL: Downlink; UL: Uplink.

**Table 4 sensors-22-06066-t004:** Weights Setup.

Test	Weights ^1^
1	[0.3 0.2 0.1 0.09 0.03 0.03 0.25]
2	[0.3 0.4 0.15 0.05 0.05 0.02 0.03]
3	[0.1 0.05 0.15 0.3 0.3 0.08 0.02]
4	[0.136, 0.144, 0.144, 0.144, 0.144, 0.144, 0.144]
5	[0.144, 0.236, 0.04, 0.334, 0.165, 0.078, 0.003]

^1^ [“Latency”, “Jitter”, “Loss”, “Bandwidth”, “Transfer”, “Distance”, “Reliability”].

**Table 5 sensors-22-06066-t005:** VIKOR Method Results.

Test	Slice 1 (%)	Slice 2 (%)	Slice 3 (%)
1	68.4	4.4	27.2
2	17.2	68.0	14.8
3	78.4	21.6	0.0
4	93.2	6.4	0.4
5	100.0	0.0	0.0

**Table 6 sensors-22-06066-t006:** COPRAS Method Results.

Test	Slice 1 (%)	Slice 2 (%)	Slice 3 (%)
1	6.0	33.6	60.4
2	14.0	25.6	60.4
3	0.0	1.6	98.4
4	1.2	10.4	88.4
5	100.0	0.0	0.0

**Table 7 sensors-22-06066-t007:** TOPSIS Method Results.

Test	Slice 1 (%)	Slice 2 (%)	Slice 3 (%)
1	10.8	64.8	24.4
2	47.2	9.2	43.6
3	0.0	0.8	99.2
4	4.8	18.4	76.8
5	100.0	0.0	0.0

**Table 8 sensors-22-06066-t008:** PrometheeII Method Results.

Test	Slice 1 (%)	Slice 2 (%)	Slice 3 (%)
1	14.4	52.4	33.2
2	19.2	24.0	56.8
3	0.0	0.4	99.6
4	9.2	20.0	70.8
5	0.8	1.2	98.0

## Data Availability

Not applicable.

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
