# Peer review of "A Novel Approach to Multi-Provider Network Slice Selector for 5G and Future Communication Systems"

_sensors, 2022, doi:10.3390/s22166066_

Round 1
Reviewer 1 Report
It is an interesting paper. It is suggested to make the words in Figure 4 larger for easy reading.
1. Please appropriately add the description of the core working details of the multi-Provider Network Slice Selector proposed in this paper. For example, how does "rankDataMethod" work when different multicriteria methods are applied?2. There are many features that affect QoS/QoE. How does the approach proposed in this paper define and evaluate "the best slice"?
3. In Algorithm 1, what are the differences between QoS and QoE, and what are their definitions?
4. The header "DataFrame" for Algorithm 1 lacks Spaces. The syntax of the two Input parameters should be the same.
5. In Algorithm 2, what does "RV/RC/RT/RP" represent respectively? What does " RV[0] " stand for? What is the data granularity of the while loop? Please describe it clearly.
6. Please give a brief introduction to the four methods (VIKOR/TOPSIS/COPRAS/Promethee II) compared in this article.
7. The spelling of "internall" in Figure4 is wrong, please check and correct.
Author Response
July 29, 2022
Dear Reviewer of Sensors,
First of all, we must say that your assessment is very important to improve our work. We believe that we have complied with all requests for review and are respectfully submitting our responses to your questions in the attached file.
Further information about our revised contribution follows in the Manuscript Status section of this submission. All changes made in the manuscript are in red text.
We keep open for any clarification and requests.
Thank you very much for your attention and consideration.
Yours Sincerely,
We, The authors

Reviewer 2 Report
A Novel Approach to Multi-Provider Network Slice Selector for 5G and Future Communication Systems: The manuscript is about improving the network slice selection with a proposed framework with cooperation between User Equipments and network edge. The topic is current, well-written and has good grammar.
The authors should consider the following suggestions provided by the reviewer to improve the scientific depth of their manuscript. They should also address the following comments to improve the quality of the presentation of their manuscript.
1- It is very hard to recognize the contribution of the paper. It should be better to represent the contributions of the paper with a separate paragraph or a subsection.
2- It is well written with good grammar. It should be a space between references, e.g., line 398 ” …geographic location [12,13,51].” ; line 394 “…be considered [12,68].”
3- The literature must be strongly updated with some relevant papers focused on the requirements of the 5G within the manuscript [1].
[1] "Waveform design considerations for 5G wireless networks." Towards 5G Wireless Networks-A Physical Layer Perspective (2016): 27-48.
Author Response

(The authors gave the same response as above.)
